# pH-Triggered Hydrogel Nanoparticles for Efficient Anticancer Drug Delivery and Bioimaging Applications

**DOI:** 10.3390/pharmaceutics16070931

**Published:** 2024-07-11

**Authors:** Keristina Wagdi K. Amin, Ágota Deák, Miklós Csanády Jr., Nikoletta Szemerédi, Diána Szabó, Árpád Turcsányi, Ditta Ungor, Gabriella Spengler, László Rovó, László Janovák

**Affiliations:** 1Department of Physical Chemistry and Materials Science, University of Szeged, Rerrich Béla tér 1, H-6720 Szeged, Hungary; keristina.wagdi@chem.u-szeged.hu (K.W.K.A.); agotadeak@chem.u-szeged.hu (Á.D.); 2Department of Chemistry, Faculty of Science, Suez Canal University, Ismailia 41522, Egypt; 3Department of Oto-Rhino-Laryngology and Head & Neck Surgery, University of Szeged, Tisza Lajos krt. 111, H-6724 Szeged, Hungary; csanady.miklos.2@med.u-szeged.hu (M.C.J.); diniklinik@freemail.hu (D.S.); office.orl@med.u-szeged.hu (L.R.); 4Department of Medical Microbiology, Albert Szent-Györgyi Medical School, University of Szeged, H-6725 Szeged, Hungary; szemeredi.nikoletta@med.u-szeged.hu (N.S.); spengler.gabriella@med.u-szeged.hu (G.S.); 5MTA-SZTE Lendület “Momentum” Noble Metal Nanostructures Research Group, University of Szeged, Rerrich B. sqr. 1, H-6720 Szeged, Hungary; tarpad@chem.u-szeged.hu (Á.T.); ungord@chem.u-szeged.hu (D.U.)

**Keywords:** polymeric nanoparticles, imaging, drug delivery, targeted release, pH-responsive

## Abstract

In this work, we developed multifunctional hydrogel nanoparticles (NPs) that can encapsulate anticancer drugs and imaging contrast agents as well. Mitomycin C (MMC) and rhodamine B (RB) were selected as models for anticancer drugs and imaging contrasting agents, respectively. Both MMC and RB were linked to the succinated polyvinyl alcohol polymer (PVA-SA). The selected labeled hydrogel NPs ((0.5% RB)-PVA-SA NPs and (1.5% RB)-PVA-SA NPs) improved the RB quantum yield from 29.8% to a minimum of 42.7%. Moreover, they showed higher emission stability compared to free RB when they were repeatedly excited at 554 nm for 2 h. Furthermore, the dye polymeric interactions significantly increased the RB fluorescence lifetime by approximately twofold. All these optical properties pave the way for our labeled hydrogel NPs to be used in imaging-guided therapy. For the labeled MMC-loaded NPs, the MMC-binding efficiency was found to be exceedingly high in all synthesized samples: a minimum of 92% was achieved. In addition, the obtained pH-dependent drug release profiles as well as the cytotoxicity evaluation demonstrated the high potential of releasing MMC under acidic cancerous conditions. Moreover, the in vitro cellular uptake experiment confirmed the accumulation of MMC NPs throughout the cytoplasm.

## 1. Introduction

Currently, a wide range of neoplasms are treated with chemotherapy, which includes the administration of one or more anticancer drugs. However, the high toxicity of anticancer drugs to normal cells and their short-term therapeutic effects reduce their efficacy. As a way to overcome these side effects, many targeted drug delivery systems (DDs) have been innovated, including inorganic nanomaterials, liposomes, and hydrogel NPs [1,2]. Polymeric materials, especially hydrogels, have received great attention over the last two decades. This can be attributed to their biodegradable and biocompatible nature as well as their long blood circulation time. Moreover, polymeric materials have high binding efficiency for a variety of organic, inorganic, and biological agents and can control drug release into tumor cells [1,3]. Among these hydrogel-forming polymers, polyvinyl alcohol (PVA) has been approved for in vivo applications and clinical use. PVA has several benefits, including its biocompatibility, hydrophilic nature, weak susceptibility to protein adsorption, and high ability to control physicochemical properties [3,4]. All these remarkable properties can be exploited to develop targeted polymeric DDs for better treatment efficacy and more controlled drug release.

However, attempts are not only focused on developing polymeric DDs but also on designing multifunctional ones that can encapsulate anticancer drugs and imaging contrast agents simultaneously within one system [5]. Considering this, a single multifunctional system containing PVA bound to both an anticancer drug and an imaging agent has been developed. As a model for anticancer drugs, mitomycin C (MMC) was selected. MMC serves as an alkylating agent that can inhibit cellular protein, DNA, and RNA synthesis. As a model for contrasting agents, the decision was to use fluorescent-based probes owing to their high sensitivity, high selectivity, real-time imaging ability, and capability for long-term tracking. The most common fluorescent imaging agents involve semiconductor nanocrystals, silica nanoparticles, and organic fluorescent dyes. However, it has been observed that organic fluorescent dyes are widely applied in cell labeling due to their ideal spectroscopic characteristics [6]. Rhodamine B (RB) is one of these organic dyes that possesses superior photophysical characteristics, including a broad emission wavelength and a relatively high quantum yield [6,7]. Therefore, RB and MMC were both conjugated to the biocompatible polymer (PVA) to design our multifunctional system.

For conjugation purposes and self-assembled hydrogel NP formation, PVA was modified to its succinylated form, as described in our previously published work [8]. There are a lot of reasons to perform this modification step. First, succinylated PVA (PVA-SA) possesses carboxyl groups that can bind to MMC through an amide bond. Since the amide bond is known to be more easily hydrolyzed in acidic conditions than in neutral ones [9], this raises the potential for targeting the release of MMC into cancer cells. Because, as we know, the extracellular pH (pH_e_) for cancer cells is lower than that of normal ones. Due to the hypoxic microenvironment in cancer cells, the pH_e_ can be as low as 5.7–6.5 [10,11]. Moreover, cancer cells’ endosomal and lysosomal environments were found to be acidic, with a pH value of 5.8 or 5.5 [12,13]. Therefore, using PVA-SA to form a pH-sensitive amide bond with MMC can enhance the specificity of cancer cell targeting. Second, the negatively charged carboxylate groups in PVA-SA can provide an opportunity for electrostatic interactions with the positively charged groups in RB molecules [14]. Besides these two reasons, PVA-SA’s carboxyl groups can create ester bonds with the existing hydroxyl ones, resulting in self-assembled hydrogel NPs [15]. All these interactions can be performed when 1-(3-Dimethylaminopropyl)-3-ethylcarbodiimide hydrochloride (EDC) is used as a crosslinker.

In the current work, we first synthesized the labeled hydrogel NPs (RB-PVA-SA NPs). After that, the focus shifted towards the one-pot synthesis of the fluorescent MMC-loaded hydrogel NPs (RB-PVA-SA-MMC NPs). The success of the conjugation reactions was confirmed using FTIR and Raman spectroscopy. The NP formation was investigated using turbidity measurements. The hydrodynamic diameter (DH), polydispersity index (PDI), and ζ-potential of the forming NPs were determined using a dynamic light scattering (DLS) device. The morphology was examined using a transmission electron microscope (TEM). Moreover, the emission properties of the synthesized fluorescent hydrogel NPs were investigated to examine their suitability for biomedical imaging. Furthermore, the in vitro drug release experiment was performed under two different pH conditions (5.8 and 7.4). Eventually, the cytotoxicity as well as the cellular uptake of RB-PVA-SA-MMC were evaluated.

## 2. Materials and Methods

The Appendix A provide a full explanation of the required chemicals needed to synthesize both succinic anhydride precursor and PVA-SA, as well as their synthetic procedures, which were performed according to our previous work [8].

### 2.1. Materials

MMC (C_15_H_18_N_4_O_5_) was acquired from 1Pluschem LLC., San Diego, CA, USA. RB (C_28_H_31_ClN_2_O_3_) and EDC (Mw 191.7, 98+%) were purchased from Sigma-Aldrich, St. Louis, MO, USA. Phosphate-buffered saline solution (PBS, pH 5.8 or 7.4) was prepared using sodium dihydrogen phosphate monohydrate (NaH_2_PO_4_.H_2_O), di-sodium hydrogen phosphate dodecahydrate (Na_2_HPO_4_.12H_2_O), and sodium chloride (NaCl), which were all purchased from Molar Chemicals Kft. Ethanol and acetone were acquired from VWR Chemicals BDH, Radnor, PA, USA. All purchased materials were utilized with no additional purification. Ultrapure water was obtained using a Millipore purification device (18.2 MΩ·cm) (Burlington, MA, USA).

### 2.2. Synthesis of RB-PVA-SA and RB-PVA-SA-MMC Hydrogel NPs by a Simple One-Pot Reaction

RB-PVA-SA hydrogel NPs were prepared using the subsequent steps: 100 mg of PVA-SA (COOH content = 2.5 mmol/g polymer) was dissolved in 10 mL of ultrapure water. Next, various RB contents (0.5, 1.0, 1.5, 3.0, and 6.0 wt%, referring to PVA-SA) were added to the prepared solutions. After that, approximately 125 mg of EDC carboxyl group activator was introduced to the reaction mixture. The resulting samples were then stirred at room temperature for almost 1 h. The self-assembled RB-PVA-SA NPs were then precipitated with an excess amount of acetone and centrifuged to remove the solvent. After that, the resulting NPs were washed several times with a slight amount of acetone or ethanol to ensure their purity. The purified NPs were then lyophilized using the freeze-dryer apparatus (Christ Alpha 1–2 LD plus). The synthesis of drug-loaded NPs (RB-PVA-SA-MMC) was executed as per the above-stated procedures. The only exception was the addition of 4 mg MMC (0.4 mg/mL) or 12 mg MMC (1.2 mg/mL) to the prepared mixtures just before EDC addition. After the synthesis and centrifugation of NPs, supernatants were collected. The absorbance was then determined using a UV spectrophotometer, either at 554 nm to determine the RB content or at 364 nm to determine the MMC content in the collected supernatants. Then, the unconjugated dye and the unconjugated drug were calculated using the RB calibration curve (Appendix A) and the MMC calibration curve in MQ water (Appendix A). Hence, Equations (1) and (2) were used to calculate the percentages of dye or drug loading as well as the binding efficiency.
(1)$$Loading\left(\%\right)=\frac{Actual\;Weight\;of\;RB\;or\;MMC\;in\;NPs}{Total\;weight\;of\;NPs\;after\;synthesis}\times 100.\;$$
(2)$$Binding\;Efficiency(\%)=\frac{Initial\;RB\;or\;MMC\;weight\;-\;RB\;or\;MMC\;weight\;in\;supernatant}{Initial\;RB\;or\;MMC\;weight}\times 100.$$

### 2.3. Macromolecular Self-Assembly and Hydrogel NP Formation

The formation of self-assembled NPs was investigated by turbidity measurements. The measurement was carried out at 25 °C using a turbidity meter LP2000 (Hanna Instruments, Service Kft, Szeged, Hungary). After adding 125 mg of EDC to two aqueous samples (10 mL), the first one containing 1.3% PVA-SA polymer without any RB and the second one containing 0.65 mg of RB, the change in turbidity was recorded over time. The experiment was performed in triplicate, and all data are expressed as the mean ± standard deviation.

### 2.4. Characterization of the Produced Hydrogel NPs

#### 2.4.1. Structural Characterization and Compositional Analysis of the Synthesized Samples

FTIR spectroscopic measurements for both the original and the prepared compounds were performed with the help of a Jasco FT/IR-4700 spectrometer (Tokyo, Japan). The spectra were recorded between 4000 and 500 cm^−1^ through the collection of 128 scans with a resolution of 1 cm^−1^. For analysis, the purified powder form of the resulting products was used.

The Raman spectra were also recorded with a Bruker Sunterra II Raman microscope (Tokyo, Japan) to prove the coupling reaction between MMC and PVA-SA. The spectra of the samples were collected by applying a near-IR light source (785 nm) tuned at 100 mW laser power. Final spectra were obtained by averaging 5 scans per measurement. For analysis, the purified powder form of the resulting products was used.

Energy-dispersive X-ray (EDX) measurements were performed using a Röntec EDS detector (Watford, UK) at 10 keV for elemental composition analysis of PVA-SA and conjugated forms. The morphology was examined by field emission scanning electron microscopy (SEM Hitachi S-4700 microscope, Tokyo, Japan) with a 10 kV acceleration voltage. Dried powder samples were used for this measurement.

#### 2.4.2. Particle Size, ζ-Potential, and Morphology Characterization

The investigation of DH, PDI, and ζ-potential values was carried out utilizing a Horiba SZ-100 Nanoparticle Size Analyzer (Kyoto, Japan). The analysis was performed at 25 °C with a 90° scattering angle. Purified and lyophilized NPs were first dispersed in PBS (pH 5.8 or 7.4, 0.9% NaCl). Then, sonication was applied for 10 min to achieve homogenous dispersion. The morphology and NP size were also investigated using a Jeol (Tokyo, Japan) JEM-1400plus transmission electron microscope (with an acceleration voltage of 200 kV). For TEM analysis, the diluted colloid samples were dropped on carbon film-covered copper grids and then allowed to dry overnight. Later, TEM images were acquired and then analyzed using ImageJ 1.53t software.

#### 2.4.3. Fluorescence Emission and Decay Properties of RB-PVA-SA Hydrogel NPs

The emission of NPs was checked using a Leica DM IL LED (Wetzlar, Germany) inverted laboratory fluorescence microscope. For the microscopic study, a dispersion of RB-PVA-SA NPs (C_RB_ = 1.5 µg/mL) was prepared. The sample was then dropped on a glass slide and examined. Moreover, the maximum emission bands and the photostability of the free and coupled RB were investigated. For the photostability measurements, RB and RB-PVA-SA samples were first prepared, each containing RB at a concentration of 1.5 µg/mL. Following this, samples were repeatedly excited at 554 nm for 2 h using a JASCO FP-8500 fluorimeter equipped with a Xe lamp light source. To assess the fluorescence intensity stability, the emissions were observed at the peak maxima after each excitation. The measurements were performed with a 4-sided quartz cuvette of 1 cm optical length. The spectra were recorded using a 2.5–2.5 nm bandwidth, a scanning speed of 200 nm/min, and a resolution of 1 nm. The absolute quantum yield was also determined by analyzing the incident light spectra and the indirect and direct excitation spectra of the samples. For the measurements, the same JASCO fluorimeter was used, equipped with the JASCOILF-835 integrating sphere (d = 100 mm). For the calibration of the instrument, a calibrated JASCOESC-842 WI lamp was utilized; thus, no other references were required. For the quantum yield calculations, the Spectra Manager 2.0 software of the apparatus was applied. The time-correlated single-photon counting method was also used to figure out the emission lifetime (ns) of both the free RB and the RB-loaded PVA-SA NPs. The measurement was conducted using a Horiba Delta Flex fluorometer. A delta-diode pulsed laser operating at a wavelength of 467 nm served as an excitation light source. The emissions were detected at a wavelength of 580 nm. During the measurement, a 1 nm bandwidth was chosen, and 10,000 counts were recorded on the peak channel. The instrument response function (IRF) was generated using a standard LUDOX^®^ colloidal silica dispersion (Grace, Columbia, MD, USA). To determine the essential lifetime components, a two-exponential fit was applied to the resulting curves, and Chi-square (ꭓ^2^) values were used to assess the goodness of the fit.

### 2.5. In Vitro Release of MMC from Resulting Hydrogel NPs

The in vitro drug release study was conducted on free MMC, ((0.5% RB)-PVA-SA)-MMC NPs, and ((1.5% RB)-PVA-SA)-MMC NPs. The unconjugated MMC was used in its purchased powder form, while the conjugated ones were used in their purified lyophilized state. The study was performed in PBS medium (pH = 5.8 or 7.4) at 37 °C over 221 h. The powder samples with 2 mL of PBS were inserted into dialysis membranes (MWCO 14,000 Da) and then immersed in 48 mL of PBS. At certain time intervals, 3 mL was withdrawn from the dissolution medium to measure spectrophotometrically the absorbance value at a wavelength of 364 nm. Then, the extracted volume was reintroduced into the release medium. The released MMC concentration was then calculated using the calibration curves presented in Appendix A. Following that, the MMC release percentage was calculated by dividing the released concentrations at each time point by the maximal concentration (0.01 mg/mL). These release experiments were repeated in triplicate. Moreover, the kinetics of the resulting release profiles were evaluated via various kinds of mathematical models (zero- and first-order models, as well as Higuchi–, Hixson–Crowell– and Korsmeyer–Peppas models) [16].

### 2.6. Cell Lines and Cell Culture

To evaluate the biological effects of the compounds, human cancer and normal cells were also used. For cancer cells, the HEp-2 larynx epidermoid carcinoma cells were used. However, for normal cells, the CCD-19Lu human lung fibroblast cells were utilized. The cell lines were originally obtained from the American Type Culture Collection (ATCC). All cell lines were cultured in Eagle’s Minimal Essential Medium (EMEM), containing 4.5 g/L glucose, vitamins, a non-essential amino acid mixture, and 10% heat-inactivated fetal bovine serum. Cells were cultured in a 37 °C incubator under standard conditions at 95% humidity and 5% CO_2_. For the experiments, the pH of the culture medium was adjusted to reach 7.4 or 5.8 through the addition of hydrochloric acid solution.

### 2.7. In Vitro Cytotoxicity Assays

The MTT [3-(4,5-dimethylthiazol-2-yl)-2,5-diphenyl tetrazolium bromide] assay was used to evaluate the cytotoxicity of RB-PVA-SA, RB-PVA-SA-MMC, and MMC. To perform this assay, normal CCD-19Lu cells and cancer HEp-2 cells were used. The adherent normal and cancer cells were cultured in 96-well flat-bottomed microtiter plates using EMEM. The density of the cells was adjusted to 6 × 10^3^ cells in 100 µL per well. The cells were seeded for 24 h at 37 °C with 5% CO_2_ before the assay. After 24 h, the medium was removed from the plates, and 100 µL of fresh medium (with the desired pH value) was added. Except for the medium control wells, samples were diluted in 100 µL of the medium and added to the cells in each well after dilution. The final concentrations for the labeled carrier (1.5% RB-PVA-SA NPs) were about 1, 2, 4, 8, 16, 31, 63, 125, and 250 µg/mL. However, for the free MMC and the MMC-loaded NPs, the MMC final concentrations were 0.2, 0.4, 0.8, 1.6, 3, 6, 13, and 25 µg/mL. Following a 72 h incubation period at 37 °C, 20 µL of the 5 mg/mL MTT stock solution (Sigma-Aldrich) was added to each well. After a further incubation at 37 °C for 4 h, 100 µL of a 10% sodium dodecyl sulfate solution (Sigma-Aldrich) was added to each well, and the plates were further incubated at 37 °C overnight. Cell growth was determined by measuring the optical density (OD) at 540 nm (ref. 630 nm). This was achieved using a Multiscan EX ELISA reader (Thermo Lab Systems, Cheshire, WA, USA). The mortality percentage was then calculated according to Equation (3). Moreover, the inhibitory concentration (IC_50_) was determined by applying the nonlinear regression dose–response curve fitting using GraphPad Prism 10.0.2. software. Equation (4) represents the fitting equation used by the software for the IC_50_ calculation.
(3)$$Mortality\%=100-\left[\frac{OD\;sample\;-\;OD\;medium\;control}{OD\;cell\;control\;-\;OD\;medium\;control}\right]\times 100.$$
(4)$$Normalized\;response=\frac{100}{1+10^{\left((\log IC50-\log inhibitor\;concentration)\times Hill\;slope\right)}}$$

### 2.8. In Vitro Cellular Uptake of RB-PVA-SA-MMC Hydrogel NPs

HEp-2 carcinoma cells (2 × 10^5^ cells/well) were cultured at 37 °C in a humidified atmosphere containing 5% CO_2_ for 24 h. Cells were then incubated with ((0.5% RB)-PVA-SA)-MMC NPs or ((1.5% RB)-PVA-SA)-MMC NPs at a concentration of 100 μg/mL at 37 °C for 12 h. Following the incubation period, cells were washed and stained with BioTracker 488 Green Nuclear Dye for 20 min at room temperature. After that, cells were rinsed before adding the fresh medium. Finally, a Leica DM IL LED inverted laboratory fluorescence microscope was used to investigate the cellular uptake of NPs.

### 2.9. Statistical Analysis of Results

Data were statistically analyzed using a one-way ANOVA or 2-sample *t*-test. A one-way ANOVA, followed by either Tukey’s or Dunnett’s HSD post hoc test, was used when comparing multiple groups. However, the 2-sample *t*-test was used when only two groups were included in the analysis. The following designation was used for the level of significance: * *p* < 0.05; ** *p* < 0.01; *** *p* < 0.001.

## 3. Results

### 3.1. Structural Characterization and Compositional Analysis of the Synthesized Samples

The introduction of carboxylic groups into the polymeric chain has been achieved by converting PVA into PVA-SA, as described in detail in the Appendix A. By applying a simple acid–base titration method, the amount of carboxylic groups in PVA-SA could be determined to be 2.5 mmol/g. It represents almost 16% of the available OH groups (16 mmol/g, as determined using acetic anhydride/pyridine titration [3]). These inserted carboxylic groups enabled PVA-SA to bind to both MMC (through an amide bond) and RB (through an electrostatic bond) [8,14]. Moreover, the uncoupled carboxylic groups can form an ester bond with the OH pendant groups in PVA-SA. With the help of the EDC crosslinker, this ester bond can be formed, leading to a self-assembled hydrogel NP, as illustrated in Figure 1.

The FTIR analysis verified the effective modification of the initial PVA as well as the conjugation reactions (Figure 2). The initial PVA spectrum showed distinctive peaks at 3275 cm^−1^ for the broad OH group and at 2934 and 2910 cm^−1^ for CH_2_ stretching vibrations. Moreover, the peak noticed at 1725 cm^−1^ is characteristic of C=O stretching vibrations. However, the bands observed at 1425, 1370, and 1318 cm^−1^ are attributed to CH and CH_2_ bending vibrations. Meanwhile, the absorption bands at 1238, 1080, and 833 cm^−1^ can be assigned to the stretching vibrations of C–O–C, C–O and C–C bonds, respectively [8]. The formation of succinylated PVA was confirmed after observing a shift in the OH vibrational band and a lowering in its intensity. Moreover, the emergence of two additional absorption bands provided further confirmation. The first band was observed at 1660 cm^−1^, and it is attributed to the carboxylate moiety. However, the second band, which appeared at 1160 cm^−1^, is due to the vibrations of the C–O bonds existing in the newly formed ester groups [17,18,19]. One could see that the carboxylate moiety band (1660 cm^−1^) had much weaker intensity in the case of NPs compared to PVA-SA. This decrease in intensity proved the ester bond formation between the available COOH and OH groups as well as the particle formation [3]. The synthesis of RB-loaded polymeric NPs was also verified through the evaluation of the resulting IR spectrum. There was a shift in the OH vibrational band from 3392 to 3318 cm^−1^, which confirmed the hydrogen bond formation between RB and PVA-SA [20]. There was also a shift in the C=O band from 1722 to 1705 cm^−1^, which confirmed the electrostatic interaction between RB and PVA-SA. Furthermore, the RB-PVA-SA spectrum showed absorption bands in the region between 1580 and 1470 cm^−1^. These bands were similar to those present in the original RB spectrum and can be assigned to the aromatic ring stretch [21,22]. Moreover, the band corresponding to the NH stretching vibration was observed at 1644 cm^−1^ in the RB spectrum and 1635 cm^−1^ in the RB-PVA-SA spectrum. This shift can be attributed to the electrostatic interaction between RB and PVA-SA [21]. For the drug-loaded spectrum, the C=O amide band and the C–N stretch band appeared at 1720 cm^−1^ and 1360 cm^−1^, respectively. These bands confirmed the presence of amide bonds between the carboxylate groups of PVA-SA and the NH groups of MMC [8,23].

Raman spectroscopy was also used to provide additional evidence for the conjugation of the MMC to PVA-SA. Appendix A shows all the registered Raman spectra. Compared to the PVA-SA spectrum, the PVA SA-MMC spectrum included a new band at 1011 cm^−1^. This band is similar to that present in the MMC spectrum and can be assigned to the C–N stretching vibration. Furthermore, the formation of the amide bonds between the carboxylate groups of PVA-SA and the NH groups of MMC was also confirmed by observing the specific amide bands in the PVA SA MMC spectrum. The C=O amide I bands were detected at 1660 and 1717 cm^−1^ [24,25]. However, the C–N amide II bands were observed at 1570 and 1609 cm^−1^, while the C–N amide III bands were noted at 1327 and 1366 cm^−1^ [24,25,26].

EDX measurements were also used to detect the presence of elements in the different conjugated forms. Compared to PVA-SA, which does not have nitrogen, RB-PVA-SA and RB-PVA-SA-MMC NPs exhibit the nitrogen element (Appendix A). These findings support the conjugation reaction of RB and MMC to PVA-SA.

### 3.2. Self-Assembly of Polymeric Chains and Formation of RB-PVA-SA Hydrogel NPs

To investigate the polymer self-assembly, two polymeric aqueous solutions were prepared. Each of them has a PVA-SA concentration of 1.3 wt/v%. However, the second one also contains 0.5 wt% of RB (referred to PVA-SA). After EDC addition, the turbidity change was recorded over time, and the results can be seen in Figure 3A. A turbidity value of about 15 NTU was recorded for both macromolecular samples within the first minute of starting the measurement. This increase in turbidity can be attributed to the carboxyl group activation, which was initiated by EDC. The turbidity reached its plateau after about 10 min of EDC addition, showing a turbidity value of 95 NTU and 89 NTU for PVA-SA and RB-PVA-SA, respectively. The noted increase in turbidity values provides evidence that self-assembled NPs were formed through ester bond formation. It can also be observed that conjugating PVA-SA to RB does not affect its self-particle formation ability. 

TEM was also used to examine the size and morphology of both PVA-SA and RB-PVA-SA NPs. The recorded pictures (Figure 3B) showed the spherical morphology of both samples. It was also feasible to determine the particle size from the images, which showed an average value of 12 ± 3 nm for PVA-SA NPs and 49 ± 15 nm for RB-PVA-SA NPs. Based on the abovementioned findings, it can be affirmed that the self-assembled polymeric nanoparticles have indeed formed.

### 3.3. Maximization of RB Content in the Hydrogel Composite Particles

Next, increasing concentrations of RB (0.5, 1.0, 1.5, 3.0, and 6.0 wt%) were employed to investigate the possibility of incorporating higher amounts of RB into the system. After quantifying the RB content present in the supernatants, the binding efficiency could be determined. As can be seen from Figure 4A, increasing the initially added amounts of RB from 0.5 to 6.0% significantly increases both the binding efficiency (from 27.35% to 40.37%) and the loading percentage (from 0.13% to 2.42%) of RB as well. 

Moreover, the size and the zeta potential of the produced particles were also checked at pH 7.4 to determine how RB maximization influences them. Figure 4B shows that the DH considerably increased from 80 ± 13 nm to 278 ± 28 nm as the nominal RB concentration changed from 0.5% to 6.0% in the NPs. Furthermore, the PDI values (Figure 4C) varied between 0.267 ± 0.10 and 0.279 ± 0.09, indicating size homogeneity for all formulations [27]. It was also observed that the zeta potential increased from −28.5 ± 2.6 mV to −60.7 ± 2.5 mV as the amount of RB increased (Figure 4D). This is because RB is not only a cationic fluorescent dye but also possesses carboxylic groups, which exist in their completely deprotonated form above a pH of 4.2. Therefore, by increasing RB loading, more ionized carboxylate groups will be present in the system, and this will lead to a higher zeta value [28,29]. The size and ζ-potential of the produced particles were also checked at pH 5.8 (Appendix A). The results were very similar to those obtained at pH 7.4 (Figure 4). 

These findings validated our polymeric NPs’ capacity to incorporate larger amounts of RB. However, for optimal drug delivery and cell internalization, it is always recommended in the literature to use NPs below 150 nm, as they can show effective tissue penetration and high cellular uptake [30,31]. As a result, the formulations (0.5% RB)-PVA-SA NPs and (1.5% RB)-PVA-SA NPs were selected to complete our further studies.

### 3.4. Fluorescence Emission and Decay Properties of RB-PVA-SA Hydrogel NPs

To ascertain whether the labeled polymeric NPs are suitable for imaging applications or not, their emission properties must be evaluated. First, a fluorescence microscope was used to check the emission ability of the labeled polymeric particles, and the recorded image (Figure 5A) confirmed this ability. According to the spectrofluorimetric data, the RB-loaded NPs experienced a red shift (λ_max_ = 589 nm) in the maximal emission band (Figure 5B) compared to the free dye (λ_max_ = 576 nm). This observed shift confirms the binding between the polymer particles and the dye molecules, since it was previously demonstrated that the red shift happens when rhodamine B is in its bound state [32,33]. Our RB-loaded NPs also showed higher emission stability compared to the free RB when exposed to continuous excitation for 2 h. As we can see in Figure 5C, the emission intensity of NPs remained almost constant; however, the emission intensity of free RB decreased to 78% after 120 min. Moreover, the produced NPs exhibited not only high stability but also high quantum yield (Figure 5D). The quantum yield of RB was found to be 29.8%, and this matches the literature value [34]. However, the quantum yields of (0.5% RB)-PVA-SA and (1.5% RB)-PVA-SA NPs were found to be 43.7% and 42.7%, respectively. This notable rise can be attributed to the role of polymers in wrapping the fluorescence centers and decreasing the motion freedoms [35].

Furthermore, the decay curves of free RB, (0.5% RB)-PVA-SA NPs, and (1.5% RB)-PVA-SA NPs can be seen in Figure 6A,B and Appendix A, respectively. The fitting of all curves resulted in two fundamental lifetime components; their values as well as their ratios are presented in Figure 6C. The shorter lifetime (τ_1_) is attributed to the presence of RB dimeric species, whereas the longer lifetime (τ_2_) is related to the presence of RB monomers [36]. For both the free RB and conjugated RB forms, the most prevalent components are the ones with the longest lifetime, accounting for more than 80% of the overall emission. However, RB-loaded NPs exhibited at least a 2.4-fold rise for τ_1_ and a 1.7-fold rise for τ_2_. This enhancement can be obviously attributed to the presence of a polymeric carrier that traps dye molecules, keeping them in a highly dispersed state [6]. This agrees with previous studies, which state that free RB molecules in solution can readily collide with one another, resulting in poor isolation and thus shortening the lifetime. However, encapsulating them in a polymeric matrix can enhance their dispersion and, hence, their decay properties [6,36,37].

### 3.5. One-Pot Synthesis of Fluorescent MMC-Loaded Hydrogel NPs

Two different concentrations of MMC (0.4 mg/mL or 1.2 mg/mL) were applied to prepare fluorescent drug-loaded NPs in a single pot. The first concentration was selected based on the recommendations outlined in several previous publications, which have demonstrated that employing this concentration results in favorable therapeutic effects [38,39]. The second one was chosen just to examine the capability of producing NPs with a higher MMC content. Figure 7 shows the results of applying the lower MMC concentration (0.4 mg/mL) in the presence of the two RB contents (0.5 and 1.5 wt%). 

As shown in Figure 7A, the MMC-binding efficiency was found to be exceptionally high, approximately 96.92% in the case of using 0.5% RB and 92.22% when using 1.5% RB. The binding efficiency of RB was also recalculated to determine whether the presence of MMC altered the previously determined values or not. As seen, the percentages obtained were nearly identical to those presented in Figure 4A, indicating that MMC has no effect on RB-encapsulated amounts.

Moreover, DLS data at pH 7.4 (Figure 7B) or at pH 5.8 (Appendix A) demonstrated that drug-loaded samples had higher mean DH values than unloaded ones. For instance, at pH 7.4, the mean DH value was about 118 ± 30 nm for ((0.5% RB)-PVA-SA)-MMC and 164 ± 29 nm for ((1.5% RB)-PVA-SA)-MMC. This finding supports the success of MMC binding. However, the ζ-potential readings of the MMC-loaded samples were found to be very similar to those of the MMC-free samples. This could imply that the majority of the encapsulated MMC is located within the NP core rather than on the surface. The spherical morphology of the MMC-loaded NPs was also confirmed via TEM measurements (Figure 7C,D). The sizes calculated from TEM images were found to be 103 ± 40 nm and 141± 39 nm for ((0.5% RB)-PVA-SA)-MMC and ((1.5% RB)-PVA-SA)-MMC, respectively. The results for the second applied MMC concentration (1.2 mg/mL) can be seen in Appendix A. For both RB contents, the MMC encapsulation percentage was found to be higher than 94%. These results demonstrate that the fluorescent composite NPs are capable of binding not only to the therapeutic concentration but also to higher concentrations.

### 3.6. In Vitro MMC Release Study

Figure 8 displays the free MMC and the conjugated MMC release profiles. The dissolution experiments were performed at physiological body temperature (37 °C) and in PBS under two different pH conditions. The first pH condition (pH 7.4) simulates the normal physiological environment. However, the second applied pH (pH 5.8) mimics the extracellular environment of cancer cells [10,11,40] as well as the endosomal and lysosomal environments [12]. 

It was clear that the conjugated MMC samples had slower release profiles compared to the free MMC, which had faster release profiles in both pH environments. For instance, it took only 30 h for free MMC to reach its maximum release percentage (90%) at pH 7.4. However, it took about 221 h for the two conjugated MMC samples to reach their maximum release percentage under the same pH condition. The reason for this prolonged release could be attributed to the presence of the amide bonds between the MMC and polymeric NPs, which are known for their very slow hydrolysis rate [9,41]. 

It can also be noticed that both the ((0.5% RB)-PVA-SA)-MMC and the ((1.5% RB)-PVA-SA)-MMC formulations showed obvious MMC release behavior in acidic conditions. For instance, after 221 h, the MMC release was found to be about 79% for the NP containing 0.5% RB and 62% for the NP containing 1.5% RB. However, after the same time span in the pH 7.4 medium, the released amount was roughly 50% for the NP containing 0.5% RB and 20% for the one containing 1.5% RB. This result was expected due to the greater susceptibility of amide bonds to hydrolysis in acidic conditions versus neutral ones [9]. 

Also, it is easy to see that the ((0.5% RB)-PVA-SA)-MMC NPs exhibit a more rapid release profile with greater cumulative release in both pH mediums compared to that of the ((1.5% RB)-PVA-SA)-MMC NPs. The reason behind these findings can be attributed to the NP sizes as well as their ζ-potential values. The MMC located within the smaller composite particles (((0.5% RB)-PVA-SA)-MMC) would be very close to the NP surface, resulting in rapid dissolution. However, NPs with larger sizes (((1.5% RB)-PVA-SA)-MMC) have large cores, resulting in much slower MMC diffusion. Moreover, since the NPs with larger sizes also have higher zeta potential values, as stated previously, this indicates that they have much higher stability compared to the smaller ones. This high level of stability could also be a reason for slowing the MMC release [42,43].

The release curves were also fitted using a set of kinetic models for the purpose of understanding the release mechanism. According to Appendix A, the Korsmeyer–Peppas model provided the best fit for all conjugated MMC release data. Since the release exponent (*n*) values for all conjugated MMC forms fall between 0.43 and 0.85, it may be concluded that a non-Fickian diffusion mechanism predominates. This means that the release mechanism depends on both the amide bond hydrolysis process and the MMC diffusion process [16]. Based on these results, it can be inferred that our polymeric DDs have the potential to slow down MMC release. Moreover, the presence of the amide bond made it possible to have a pH-dependent release and thus facilitated the release of MMC, specifically into cancer cells. Additionally, the NP characteristics can control how much drug is released and how fast it is released.

### 3.7. In Vitro Cytotoxicity

#### 3.7.1. Cytotoxicity of Labeled Polymeric NPs (RB-PVA-SA NPs)

The biocompatibility of the produced fluorescent polymeric NPs is essential for biological applications. First, (1.5% RB)-PVA-SA was selected as an example to assess the cytotoxicity towards normal human lung fibroblast cells (CCD-19Lu). Figure 9 summarizes the CCD-19Lu cell viability after being incubated for 72 h with the NPs at various doses (1, 2, 4, 8, 16, 31, 63, 125, and 250 µg/mL). As demonstrated in Figure 9, there was no obvious change to the naked eye between the untreated cells and the treated cells. More precisely, a cell viability of about 80% was detected after incubation with NPs at a concentration of 250 µg/mL for 72 h. According to the literature, compounds that exhibit about 80% cell viability are considered to have an extremely low cytotoxicity profile [44]. These results confirmed the biocompatibility of our labeled NPs, showing a high potential for effective and safe bioimaging applications.

#### 3.7.2. Cytotoxicity of MMC-Loaded Hydrogel NPs

The cytotoxicity study of free MMC and MMC-loaded NPs was performed on Hep-2 cells. The cells were incubated for 72 h with increasing concentrations of MMC in two different pH environments (7.4 and 5.8). The mortality% and IC_50_ values for the tested samples are displayed in Figure 10A,B, respectively. 

At a pH of 7.4, cell mortality was found to be much lower in the case of MMC-loaded NPs than in free MMC at all applied dosages. For instance, at a concentration of 25 µg/mL MMC, 93.3% of cells died with the free MMC, 63.9% with ((0.5% RB)-PVA-SA)-MMC NPs), and 54.0% with ((1.5% RB)-PVA-SA)-MMC NPs. This was expected because of the prodrug nature and the low accumulative release of MMC from the NPs at pH 7.4. But when the pH of the culture medium was adjusted to 5.8, which is close to the pH_e_ of most cancer cells, the percentage of cell death rose dramatically in the case of NPs at all MMC dosages. In other words, at a MMC dosage of 25 µg/mL, the mortality percentage became 90.2% for free MMC, 88.8% for ((0.5% RB)-PVA-SA)-MMC NPs, and 88.7% for ((1.5% RB)-PVA-SA)-MMC NPs. It was also noticed that the IC_50_ values for both NPs were significantly lower at pH 5.8 than they were at pH 7.4 (Figure 10B). The IC_50_ values were 0.6 ± 0.1, 6.5 ± 0.8, and 18.4 ± 3.6 µg/mL at pH 7.4 and 0.5 ± 0.2, 0.6 ± 0.1, and 0.7 ± 0.1 µg/mL at pH 5.8 for free MMC, ((0.5% RB)-PVA-SA)-MMC NPs, and ((1.5% RB)-PVA-SA)-MMC NPs, respectively. According to these findings, a significantly higher concentration of NPs is needed to cause 50% mortality in cells with a pH_e_ of 7.4 compared to cells with a pH_e_ of 5.8, which is not the case for free MMC. In other words, these results validated NPs’ capacity to exhibit more toxicity towards cancer cells, which are known for their lower pH_e_ relative to normal cells.

### 3.8. In Vitro Cellular Uptake of RB-PVA-SA-MMC Hydrogel NPs

Fluorescence microscopy was utilized to track the cellular uptake of RB-PVA-SA-MMC NPs by HEp-2 cancer cells to assess their subcellular localization. The cells were incubated with ((0.5% RB)-PVA-SA)-MMC or ((1.5% RB)-PVA-SA)-MMC NPs (red channel) at a concentration of 100 µg/mL for 12 h and successively with BioTracker 488 Nuclear Dye (green channel) for selective staining of the nucleus. The suitability of NPs for cell imaging was demonstrated by the observation of a strong red fluorescence signal, even at a comparatively low nominal content of RB (0.5 wt% of PVA-SA).

Moreover, the merged images (Figure 11) confirmed the cellular internalization of NPs within HEp-2 cells as well as their accumulation throughout perinuclear areas and the cellular cytoplasm. It was expected that NPs would accumulate highly in the cytoplasm rather than the nucleus because nuclear membrane pores are known to allow the passage of particles smaller than 9 nm [45]. Overall, the results demonstrated the great promise of both NPs to be used in cellular imaging.

## 4. Discussion

In this work, we developed fluorescent hydrogel NPs (RB-PVA-SA NPs) and MMC-loaded fluorescent polymeric NPs (RB-PVA-SA-MMC NPs). A simple and new methodology was used to design these hydrogel NPs, requiring only a few synthetic steps. During the synthesis of RB-PVA-SA hydrogel NPs, different amounts of RB were added to the system. The results showed that increasing the initially added amounts of RB from 0.5 to 6.0 wt% significantly increased both the binding efficiency (from 27.35% to 40.37%) and the loading percentage (from 0.13% to 2.42%) of RB. Furthermore, the DH considerably increased from about 80 nm to 278 nm, and the ζ-potential increased from about −28 mV to −60.7 mV. These findings validated our hydrogel NPs’ capacity to incorporate various amounts of RB. However, to work with NPs smaller than 150 nm, the two formulations (0.5% RB)-PVA-SA NPs and (1.5% RB)-PVA SA NPs were chosen for further investigations. These hydrogel NPs exhibited a high level of photostability compared to free RB when they were continuously excited at 554 nm for a duration of 2 h. Moreover, thanks to these polymeric NPs, the RB quantum yield increased from 29.8% to a minimum of 42.7%. Furthermore, the dye–polymer interactions brought about a twofold increase in the RB fluorescence lifetime. All these features made it possible for our labeled hydrogel NPs to be used in biomedical imaging. RB-PVA-SA-MMC NPs were also produced using two different initial amounts of MMC (4 mg and 12 mg). The MMC-binding efficiency was found to be extremely high in all cases, with at least 92% achieved. The in vitro drug release study demonstrated the ability of the hydrogel NPs to prolong MMC release. Furthermore, it showed how the amide bond made it possible to have pH-dependent release, which in turn can facilitate the specific delivery of MMC to cancer cells. Moreover, the cytotoxicity evaluation confirmed that RB-PVA-SA-MMC NPs at pH 5.8 were more cytotoxic toward cancer cells than those at pH 7.4. Finally, the in vitro cellular hydrogel NP uptake experiment showed the great potential of the NPs to be used in imaging-guided therapy applications. All these findings together pave the way for better treatment efficacy and more controlled drug release.

## Figures and Tables

**Figure 1 pharmaceutics-16-00931-f001:**
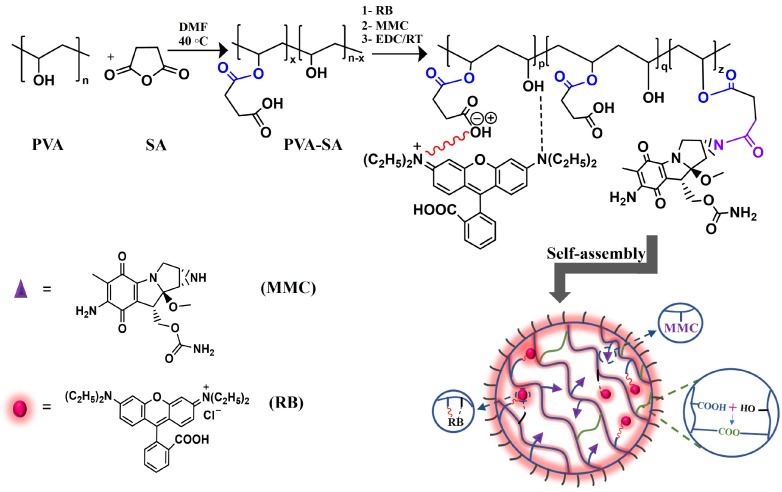
The synthesis process of succinated PVA, the coupling reactions of RB (via electrostatic interaction presented in a red zigzag line and hydrogen bond presented in a black dashed line), and MMC (through amide bonds highlighted in purple) with PVA-SA, and the crosslinking of carboxyl and hydroxyl groups to form self-assembled hydrogel NPs.

**Figure 2 pharmaceutics-16-00931-f002:**
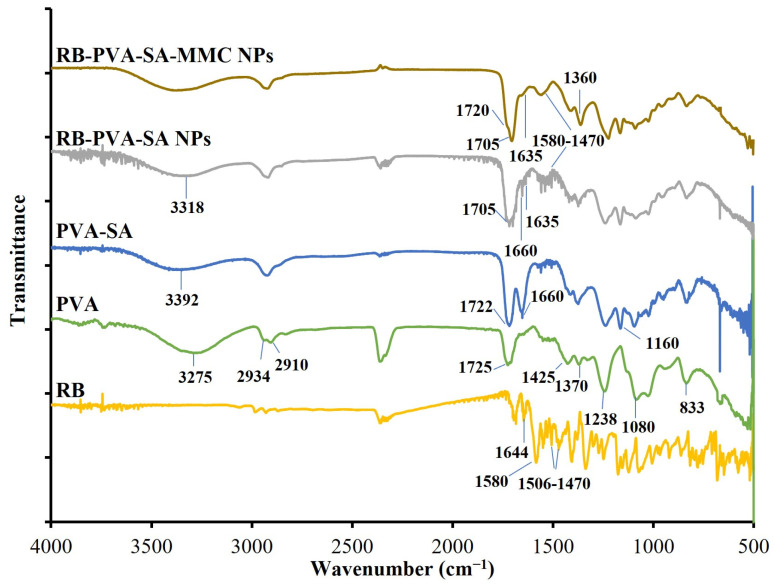
FTIR spectra of RB, PVA, and the modified PVA forms (PVA-SA, RB-PVA-SA, and RB-PVA-SA-MMC).

**Figure 3 pharmaceutics-16-00931-f003:**
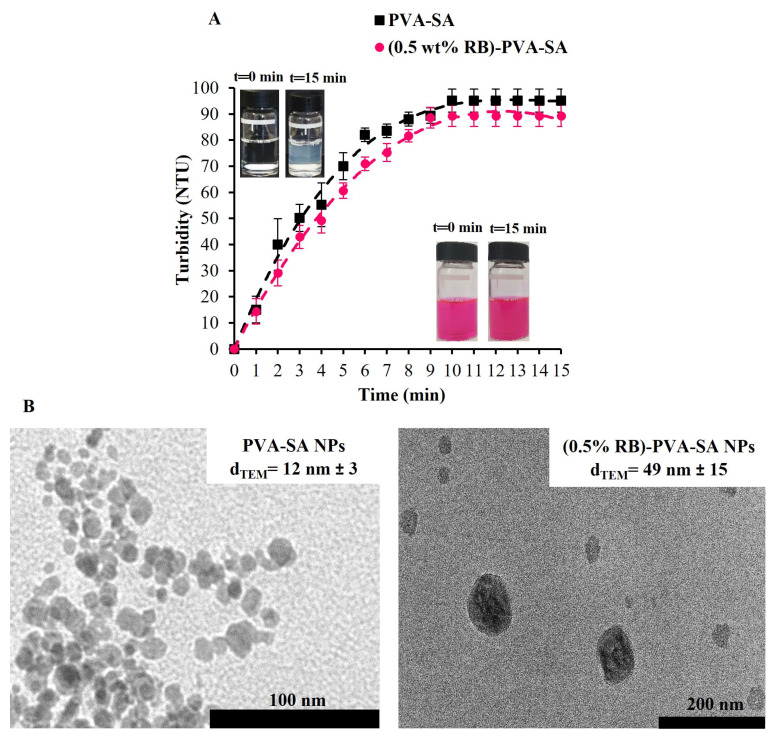
The variation in the turbidity of PVA-SA and RB-PVA-SA with EDC over time (**A**). TEM images of the produced NPs (**B**). The nominal concentration of RB in the RB-PVA-SA sample is 0.5 wt%.

**Figure 4 pharmaceutics-16-00931-f004:**
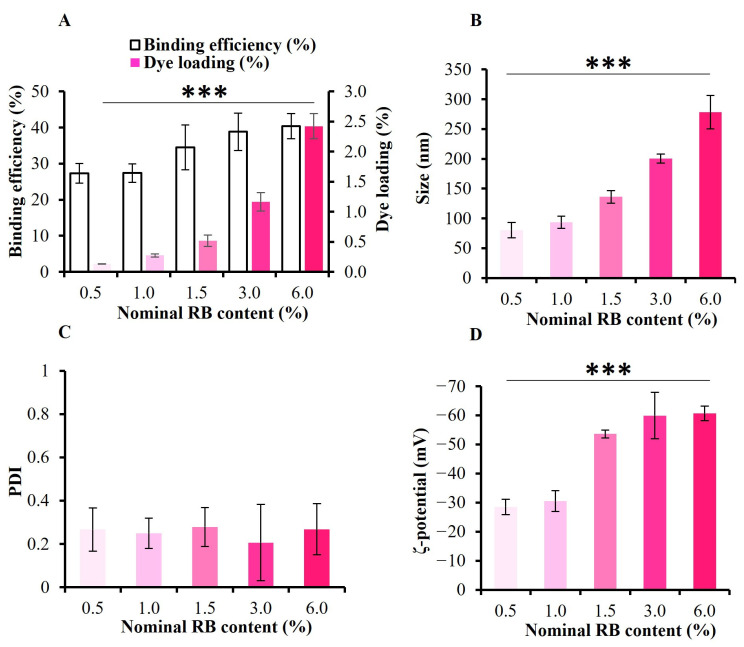
The binding efficiency results and the dye loading percentage for the prepared NPs as a function of increasing the nominal RB content (**A**). The average particle sizes (**B**), the PDI values (**C**), and the ζ-potential values (**D**) of the produced RB-loaded NPs at pH 7.4. Statistical analysis of the results: the added asterisks in the figure represent the summarized *p* value obtained from one-way ANOVA analysis. *** *p* < 0.001. The detailed outcomes of Tukey’s multiple comparisons test are presented in Appendix A.

**Figure 5 pharmaceutics-16-00931-f005:**
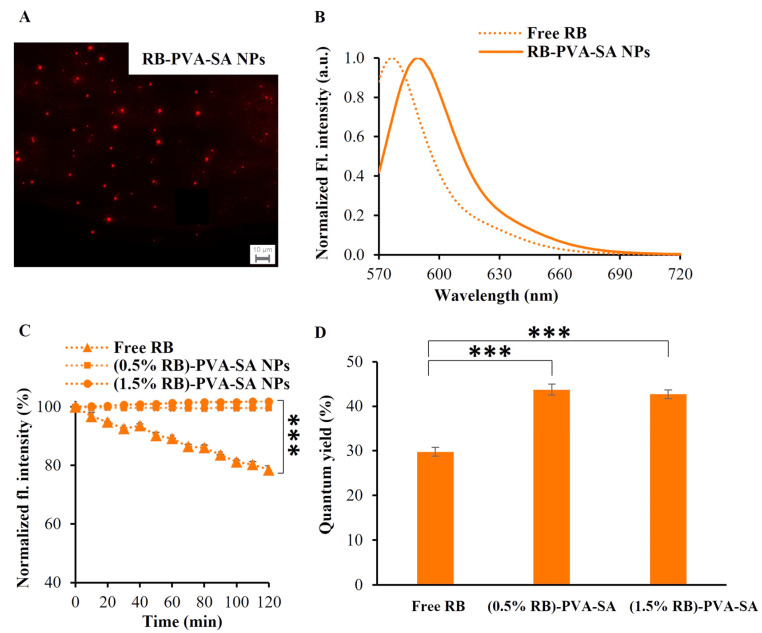
(**A**) The fluorescence microscopy image of RB-loaded NPs dispersed in PBS solution (C_RB_ = 1.5 µg/mL). (**B**) The normalized emission spectra of the free dye and the dye-loaded NPs dispersed in PBS solution. Samples were excited at their peak maxima (RB at 554 nm and RB-PVA-SA at 565 nm), and their emission was observed from 570 to 850 nm. (**C**) The fluorescence intensity stability of unbound RB and RB-loaded NPs (C_RB_ = 1.5 µg/mL) upon repeated excitation at 554 nm for 120 min using a Xe arc lamp. (**D**) The quantum yield of free RB and RB-PVA-SA NPs dispersed in MQ water. Statistical analysis of the difference in photostability and quantum yield results was performed using one-way ANOVA with Dunnett’s post hoc test (control group: free RB). *** *p* < 0.001.

**Figure 6 pharmaceutics-16-00931-f006:**
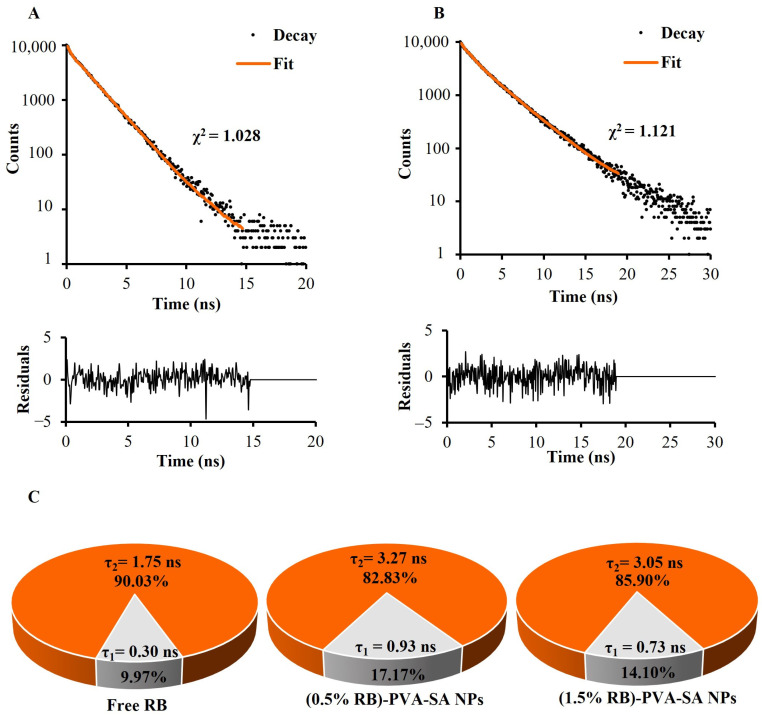
The typical fluorescence decay profiles with the fittings for free RB (**A**) and (0.5% RB)-PVA-SA NPs (**B**). The ratio of the determined two dominant lifetime components and the lifetime values (**C**) for free RB as well as (0.5% RB)-PVA-SA NPs and (1.5% RB)-PVA-SA NPs.

**Figure 7 pharmaceutics-16-00931-f007:**
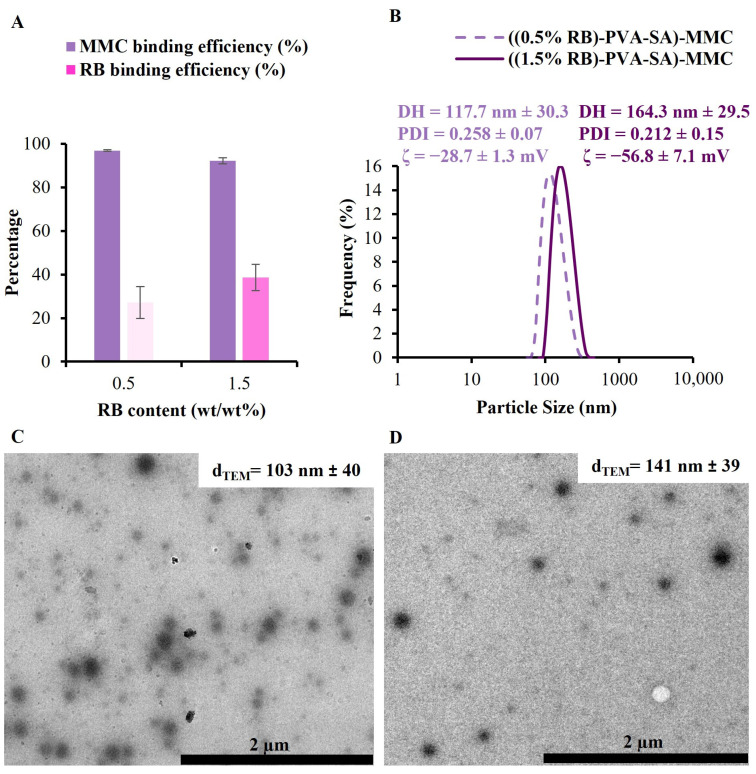
The MMC- and RB-binding efficiency results (**A**) as well as the DLS data at pH 7.4 (**B**) for the two samples obtained from applying the first MMC concentration (0.4 mg/mL) either with a 0.5% or 1.5% RB. TEM images of the produced ((0.5% RB)-PVA-SA)-MMC NPs (**C**) and ((1.5% RB)-PVA-SA)-MMC NPs (**D**).

**Figure 8 pharmaceutics-16-00931-f008:**
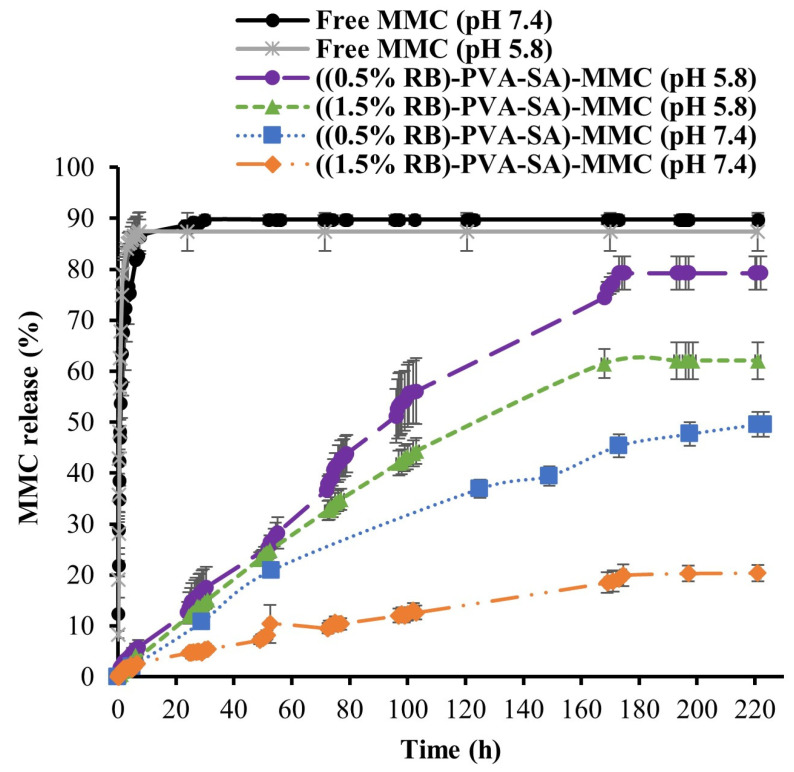
The in vitro-releasing profiles of free MMC and loaded MMC (equivalent to 0.01 mg/mL MMC) at 37 °C in PBS buffer with pH = 5.8 and 7.4.

**Figure 9 pharmaceutics-16-00931-f009:**
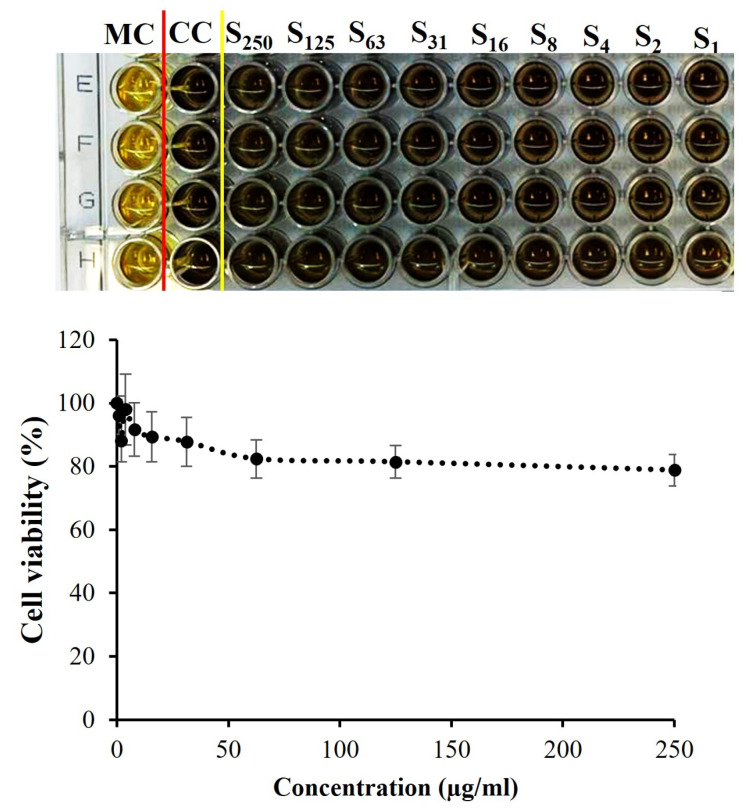
The viability of CCD-19Lu cells incubated for 72 h with (1.5% RB)-PVA-SA NPs at various concentrations. The MC is the medium control, the CC is the cell control, S is for PVASA-RB NPs, and the subscripted number indicates the NP concentration in µg/mL. The red line represents the separation line between the MC and CC wells. The yellow line represents the separation line between the CC and S wells.

**Figure 10 pharmaceutics-16-00931-f010:**
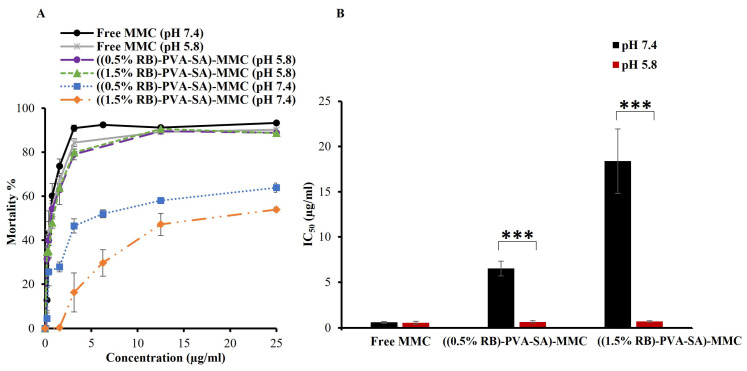
(**A**) The mortality of HEp-2 cells incubated for 72 h with free MMC, ((0.5% RB)-PVA-SA)-MMC NPs), and ((1.5% RB)-PVA-SA)-MMC NPs at varying MMC concentrations in a culture medium with a pH of 7.4 or 5.8. (**B**) The determined IC_50_ values for all samples at both pH conditions. Statistical analysis was performed using a 2-sample *t*-test. *** *p* < 0.001.

**Figure 11 pharmaceutics-16-00931-f011:**
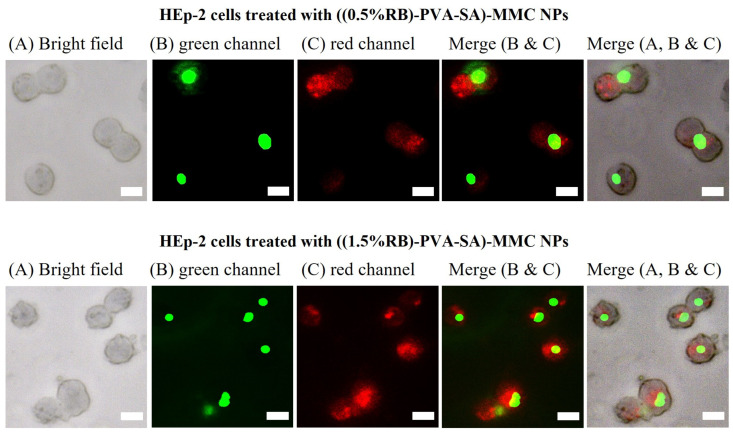
Fluorescence microscopy images of HEp-2 cancer cells after being treated with (0.5% RB)-PVA-SA-MMC NPs or (1.5% RB)-PVA-SA-MMC NPs for 12 h. The concentration of NPs is 100 μg/mL. The green fluorescence signals correspond to the BioTracker 488 Green Nuclear Dye. The red fluorescence signals correspond to the NP accumulation. The scale bar represents 10 μm.

## Data Availability

Data will be available upon request.

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
