# Peer review of "pH-Triggered Hydrogel Nanoparticles for Efficient Anticancer Drug Delivery and Bioimaging Applications"

_pharmaceutics, 2024, doi:10.3390/pharmaceutics16070931_

Round 1

Reviewer 1 Report

Comments and Suggestions for Authors

The authors have developed multifunctional hydrogel nanoparticles for dual delivery of anticancer drugs and imaging contrast agents, showing the results in drug release and stability under relevant conditions. The enhanced optical properties and high MMC binding efficiency might underscore their potential in imaging-guided therapy. I have a few comments for your consideration.

1.     Besides FTIR spectroscopy, are there any other methods to confirm the success of the conjugation reactions? Additional evidence may be required.

2.     In Figure 4, consider adding labels to indicate significant differences. Do all comparisons show significant differences?

3.     The fluorescence microscopy image of RB-loaded NPs in Figure 5A appears to show aggregation. Could this be distributed more evenly?

4.     Why do some experiments use pH 5.8 and 7.4 while others use pH 6.0 and 7.4? Could the pH levels be standardized across experiments?

5.     In the in vitro experiments, is adjusting the pH to an acidic level, such as pH 6.0, harmful to the cells themselves?

6.     Regarding Figure 9, is it necessary to obtain an IC 50? How is the IC 50 calculated when the cell viability at the highest concentration is around 80%?

Reviewer 2 Report

Comments and Suggestions for Authors

The document titled “pH-triggered hydrogel nanoparticles for efficient anticancer drug delivery and bioimaging applications” presents novel results, is properly characterized and referenced. The results may become an advance in drug-delivery systems as well as their tracking in the body. I list some doubts;

-The section 2.3 is not clear please improve the wording

-Is this correct hydrogen chloride or chloridric acid?

-Conclusion looks missing

-What is the hydrodynamic radius of the nanogels at different pH values?

-In line 566 “, and the ζ potential increased from -28.5 ± 2.6 mV to -60.7 ± 2.5 mV” is that correct?

Comments on the Quality of English Language

no comments

Round 2

Reviewer 1 Report

Comments and Suggestions for Authors

The authors addressed all of my concerns. Thanks.